# Piezo1 Channel as a Potential Target for Hindering Cardiac Fibrotic Remodeling

**DOI:** 10.3390/ijms23158065

**Published:** 2022-07-22

**Authors:** Nicoletta Braidotti, Suet Nee Chen, Carlin S. Long, Dan Cojoc, Orfeo Sbaizero

**Affiliations:** 1Department of Physics, University of Trieste, Via A. Valerio 2, 34127 Trieste, Italy; nicoletta.braidotti@phd.units.it; 2Institute of Materials, National Research Council of Italy (CNR-IOM), Area Science Park Basovizza, Strada Statale 14, Km 163,5, 34149 Trieste, Italy; cojoc@iom.cnr.it; 3CU-Cardiovascular Institute, University of Colorado Anschutz Medical Campus, 12700 East 19th Ave., Aurora, CO 80045, USA; suet.chen@cuanschutz.edu; 4Center for the Prevention of Heart and Vascular Disease, University of California, 555 Mission Bay Blvd South, Rm 352K, San Francisco, CA 94143, USA; carlin.long@ucsf.edu; 5Department of Engineering and Architecture, University of Trieste, Via A. Valerio 6/A, 34127 Trieste, Italy

**Keywords:** fibroblasts, myofibroblasts, fibrosis, heart diseases, cardiomyopathies, fibroblasts activation, Piezo1, mechanosensitive ion channel, cardiac remodeling

## Abstract

Fibrotic tissues share many common features with neoplasms where there is an increased stiffness of the extracellular matrix (ECM). In this review, we present recent discoveries related to the role of the mechanosensitive ion channel Piezo1 in several diseases, especially in regulating tumor progression, and how this can be compared with cardiac mechanobiology. Based on recent findings, Piezo1 could be upregulated in cardiac fibroblasts as a consequence of the mechanical stress and pro-inflammatory stimuli that occurs after myocardial injury, and its increased activity could be responsible for a positive feedback loop that leads to fibrosis progression. The increased Piezo1-mediated calcium flow may play an important role in cytoskeleton reorganization since it induces actin stress fibers formation, a well-known characteristic of fibroblast transdifferentiation into the activated myofibroblast. Moreover, Piezo1 activity stimulates ECM and cytokines production, which in turn promotes the phenoconversion of adjacent fibroblasts into new myofibroblasts, enhancing the invasive character. Thus, by assuming the Piezo1 involvement in the activation of intrinsic fibroblasts, recruitment of new myofibroblasts, and uncontrolled excessive ECM production, a new approach to blocking the fibrotic progression can be predicted. Therefore, targeted therapies against Piezo1 could also be beneficial for cardiac fibrosis.

## 1. Introduction

Heart failure (HF), which often follows myocardial infarction (MI), affects millions of people, with a higher frequency as the population ages [1]. Following MI, but also several other cardiac diseases (e.g., aging, hypertension, and hypertrophic cardiomyopathy [2]), fibrotic remodeling occurs with a number of undesirable outcomes, including increased hospitalization rates and an increased incidence of adverse cardiac events leading to higher mortality [3,4,5].

The cardiac fibroblast plays a pivotal role in the fibrotic process in both the reparative and reactive mechanisms [3,6,7,8]. In the first reparative and protective stage, fibroblasts act to preserve the cardiac function [9], but if long-term activation of pathological signaling is irreversibly established, the reactive mechanism emerges [7]. This not only drives a drastic change in extracellular matrix (ECM) mechanical properties, such as stiffening as a consequence of excessive ECM deposition [9,10] but also changes whole-cell mechanical properties [11,12]. These consequences are initiated by mechanotransduction processes which are well established and essential for normal heart function. Cellular mechanosensitivity is an important cell feature based on force-induced conformational changes in mechanosensitive proteins which lead to the activation of signaling pathways [13,14]. Thus, alterations in the normal cellular force transmission, as well as genetic mutations in proteins involved in downstream signaling pathways (e.g., integrins, lamins, etc.), can be directly translated into some cardiomyopathies and also other non-cardiac diseases such as muscular dystrophy [13,14]. In fact, studies based on human and mouse models of dilated cardiomyopathy linked the pathologic state to defects in muscle-cell Z-disc components involved in stretch sensing [15,16]. As well, myocytes integrin-specific loss highlighted an abnormal heart function in vivo with an increased amount of myocardial fibrosis and the development of a dilated cardiomyopathy [17].

Physical mechanotransduction relies on direct force transmission from the cell surface to the nucleus through a physical coupling between the nuclear membrane and the extracellular space by cytoskeletal components [18]. Importantly, the physical links between the cytoskeleton and nuclear membrane proteins allow the entire cell to act as a sole mechanically coupled system.

Among the aforementioned mechanosensors, stretch-activated channels (SACs) are good candidates for transducing mechanical forces into electrochemical signals [13], likely activating a variety of potential signaling pathways subsequent to their force-induced opening [14]. In the heart, several channels are expressed, and among these, the SAC Piezo1 was recently identified as a key mechanosensor regulating calcium signaling by sensing changes in membrane tension [9]. Despite several studies being carried out on a functional role for Piezo1 in cardiomyocytes and its dysfunctional role in related heart diseases [19,20], less is known about its role in cardiac fibroblasts. Some recent findings highlight the possibility of considering Piezo1 as a valid biomarker, as well as a potential target for cancer therapies [21,22]. Since the mechanical environment established in fibrosis shares some common features with the tumor paradigm, we predict that some Piezo1-dependent hidden mechanisms found effective for several cancers [23] could also be proposed for the dysfunctional regulation of Piezo1 in fibroblasts during cardiac fibrosis.

## 2. Role of the Fibroblast in Tissue Fibrosis

The myocardium is composed of cardiomyocytes and non-myocytes, which are embedded into the surrounding environment composed of proteins of the ECM [8,9,24], such as collagens, mainly type I and III, and fibronectin [25]. Cardiac fibroblasts (CFs) are the most prevalent non-myocyte cell type and account for more than half of the total cell population [8,9,24]. Cardiac fibroblasts are responsible for preserving equilibrium between ECM synthesis and degradation [26,27,28]. This balance is interrupted during inflammation, and excessive protein synthesis may occur, which is the hallmark of pathologic fibrosis [26,29]. This cardiac remodeling which occurs following MI [30], and also in hypertrophic cardiomyopathies and myocarditis [2], is an essential prerequisite for maintaining the structural integrity of the heart and preventing cardiac rupture [7], but it may lead to adverse outcomes if it acquires a chronic behavior. In fact, studies based on the prognostic impact of fibrosis implicate it in impaired diastolic function and a number of pathologic and, at times, lethal arrhythmias [2,5,31].

Under normal conditions, fibroblasts are quiescent but, thanks to their phenotypic plasticity, express the ability to adapt in response to injury [9,32,33,34]. To do so, they undergo a phenotypic transition to an activated form called a “myofibroblast” [6,35]. It is well known that the activation is a direct consequence of mechanical stress or cytokines, historically transforming growth factor β (TGF-β), associated with the inflammatory response after MI [36,37,38,39,40]. Myofibroblasts are primarily characterized by actin stress fibers, the development of which is a consequence of the cell alignment parallel to the mechanical load [41,42]. In addition, enhanced cytokines secretion and increased deposition of ECM components (mainly collagen and fibronectin) are characteristic features [41,43,44]. Alongside the increased fibronectin (FN) enriched ECM, fibroblasts are also able to lead to de novo expression of extra domain-A FN (ED-A FN), which is usually used as a marker of the early stage of phenoconversion [42,45]. ED-A FN constitutes a continuous link from ECM to integrins and stress fibers, guaranteeing the ability to exert substantial traction forces on ECM [44]. Moreover, they are characterized by the incorporation of α-smooth muscle actin (α-SMA) into the stress fibers and the development of mature focal adhesions [30,42,46].

This allows cells to acquire a higher contractile behavior and the ability to exert higher traction forces on ECM [26]. While myofibroblasts are absolutely necessary after MI, and their absence leads to inefficient scar formation and ventricular rupture [33], their persistence beyond the physiologic repair timepoint becomes pathologic for the heart. Under these circumstances, their innate characteristics, such as enhanced protein synthesis and cytokines secretion, play an adverse role and promote fibrosis. 

## 3. Paracrine and Autocrine Effectors

Communication between different cell types in the myocardium occurs through both autocrine and paracrine signaling, which are crucial under both physiological and pathophysiological conditions [47]. Paracrine signaling takes place when a cell secretes signaling molecules which influence an adjacent cell (or cells) that expresses the cognate receptor on the cell surface [47]. In the fibrotic condition, the enhanced cytokines secretion, such as transforming growth factor β (TGF-β) by myofibroblasts, acts as a paracrine signal for adjacent cells. For example, it was demonstrated that TGF-β promotes the endothelial-to-mesenchymal transition in coronary endothelial cells, representing a source for new myofibroblasts [48]. Moreover, a paracrine effect on cardiac muscle cells is also seen. In fact, TGF-β can induce alterations in myocytes’ growth and gene expression profiles [49]. It was also demonstrated that other pro-inflammatory molecules could regulate the adverse outcomes in the fibrotic heart. In fact, increased levels of Interleukins (IL), such as IL-1 and IL-6, as well as tumor necrosis factor α (TNFα), are correlated with the severity of heart failure [50,51,52,53,54]. These cytokines, secreted from infiltrating immune/inflammatory cells but also non-myocytes, especially cardiac fibroblasts, can act as paracrine effectors, as well. It was observed that the increased IL-6 secretion by fibroblasts influences cardiomyocytes negatively causing hypertrophy and reduced contractile ability [50,55,56,57].

In the autocrine context, cytokines (e.g., TGF-β, IL-1, IL-6, and TNFα) can directly influence the cells producing them. In fact, the increased TGF-β secretion is able to force the activation of adjacent quiescent fibroblasts [7,9,58,59], promoting the progression of the pathological condition. Moreover, several studies show that culturing fibroblasts in conditions in which the activated phenotype is preferred (e.g., on stiff substrate), stimulation with TGF-β is able to enhance cellular forces generation through increased incorporation of α-SMA into stress fibers [40,60,61,62]. The same happens with IL-6 [50], and this emphasizes how fast the cytokines secretion changes from a pro-inflammatory to a pro-fibrotic character with adverse effects on cardiac health [7]. Moreover, IL-1 was observed to be responsible for increased fibroblast migration, which is enhanced in combination with TNFα. This may lead to the creation of new fibrotic zones expanding the existing fibrotic tissue [52,63]. Finally, it also highlighted the role of TNFα in the regulation of other cytokines. Experiments performed on human cardiac fibroblasts revealed a TNFα-induced increase in IL-1 and IL-6 [53].

These mechanisms suggest the innate character of the fibrotic condition, which consists of the establishment of a feedback loop [35,44]. In fact, when the first fibroblast activation occurs, cytokines, as well as mechanical stresses due to the enhanced FN deposition, lead to both paracrine and autocrine pathways, which drive the recruitment of new myofibroblasts. For the reasons mentioned, the characteristics linked to this ultimate phenotype seem sufficient for its self-sustainment [41,42,43,44] and represent the central topic of the dysfunctional cardiac remodeling-related adverse outcomes (Figure 1).

## 4. Mechanical Remodeling and Paratensile Effector

The terms “stiffness” or “rigidity” denote the material property and quantify the resistance to deformation under loading. In mechanobiology, this property, described by Young’s Modulus or Elastic Modulus (measured in Pascal (Pa)), is usually used for the mechanical characterization of cells or substrates to which they adhere [11]. The continuous ECM production by myofibroblasts drives the myocardium toward a progressive increase in tissue stiffness. The initial stage, usually referred to as “reparative fibrosis”, is followed by the progressive fibrotic chronic remodeling called “reactive fibrosis” [8,9,29]. During the remodeling, the stiffness shifts from around 8 kPa, the elastic modulus of healthy myocardium, to 20–100 kPa [18,64,65]. As the stiffness increases, the activation of new fibroblasts is promoted by paratensile signaling in which the myofibroblast–fibroblast crosstalk occurs through fibrous matrix-transmitted forces [32,66]. In this mechanism, a circular loop is established where the excessive ECM-induced stiffening by myofibroblasts is responsible for the activation of new fibroblasts. Moreover, it is well demonstrated that the activation of TGF-β from the latent form is a highly integrin-dependent mechanism [67]. In fact, Henderson and co-workers observed an inhibited fibrotic character in the liver, lung, and kidney after deletion of α_v_ integrin subunit from myofibroblasts. Moreover, they addressed a potentially relevant role to the β_1_ subunit, supposing that α_v_β_1_ may be the major integrin responsible for the TGF-β activation in myofibroblasts [68]. In addition, it was demonstrated that the integrin-mediated activation requires a linkage to actin and also an active cytoskeletal actin reorganization [67,69]. This was confirmed through the increased ability of myofibroblasts to activate TGF-β via an actin–myosin dependent process thanks to the increased contractility [70]. Finally, the loop is then importantly closed if the role of TGF-β in the stimulation of ECM accumulation is considered [67,71]. Thus, it can be summarized that the stiff mechanical environment promotes the recruitment of new myofibroblasts through a paratensile but also an autocrine manner as a consequence of traction forces and contractility-dependent cytokines release, which in turn enhance the stiffness.

Therefore, the ultimate condition of this feed-forward mechanism is chronic remodeling which involves even areas that are remote from the original injury [6,72]. In addition, it was demonstrated that when cells are subjected to paratensile stimuli, calcium influx occurs and participates in actin remodeling. In fact, Longwei Liu and colleagues demonstrated the requirement of calcium in myofibroblasts transition as a consequence of paratensile signaling [66]. They observed that by blocking calcium influx, the actin outgrowth was inhibited; thus, the pivotal role of calcium in paratensile mechanotransduction was validated. It is well known that Ca^2+^ signaling occurs in cells for the regulation of several biological processes such as migration, cytoskeletal reorganization, and traction forces generation [73,74,75]. For the regulation of intracellular Ca^2+^, two main types of mechanisms are involved. Calcium influx can occur from the extracellular environment or can be initiated from intercellular compartments (e.g., endoplasmic reticulum, mitochondria, or nucleus) [73,76]. It must be emphasized that calcium transients from the extracellular side are possible thanks to mechanoreceptors located on cell membranes [33,77,78]. SACs are good candidates for cell mechanosensing which consists of the transduction of mechanical forces into a cellular electrochemical signal [13]. Even though cardiac fibroblasts are not generally perceived to be electrically excitable cells, mechanically-induced membrane potential oscillations were observed. This is possible thanks to gap junctions which mediate an electrical coupling between cardiomyocytes and fibroblasts for the propagation of the action potential across the myocardium [79,80,81], but also thanks to the presence of cation channels. 

## 5. Ion Channels in Cardiac Fibroblasts

Cardiac fibroblasts were shown to express many different SACs, both selective, such as potassium-selective channel TREK-1 [9,82], and non-selective channels, such as transient receptor potential canonical (TRPC), mainly TRPC3 and TRPC6, transient receptor potential vanilloid (TRPV), such as TRPV1 and TRPV4, transient receptor potential melastatin TRPM7, and Piezo1 [9,18,83]. For a better description of these channels within the heart, we refer the reader to [9]. This interesting work by Leander Stewart and Neil A. Turner reviews the links between some of these channels and the cardiac remodeling which occurs in fibrosis. For example, the TRPV1 channel was observed to retain a protective effect against cardiac fibrosis [84], while the TRPV4 channel gained a lot of attention in terms of both cardiac and also other organ fibrosis [85,86]. In fact, it was observed to be required for TGF-β induced differentiation of cardiac fibroblasts into myofibroblasts, also confirmed in the context of pulmonary fibrosis. However, also TRPC6 and TRPM7 were addressed to cardiac fibroblasts activation [87,88]. In addition, even cardiomyocytes expressing TRPM4 were found able to contribute in cardiac fibrosis [89]. This channel was investigated by Guo and co-workers in terms of pressure-overload induced cardiac hypertrophy regulator. In their work they observed that the fibrotic character associated with the development of pathological hypertrophy is reduced in cardiomyocytes specific TRPM4 knock-out. Finally, the role of selective potassium channel TREK-1 was also highlighted. Abram et al. [82] observed a reduction in cardiac fibrosis and fibrosis gene expression in global TREK-1 knock-out mice, marked by a reduction in fibroblasts migration and proliferation. In addition, by cell specific channel deletion, they observed that the cardioprotective effect was due to cardiac fibroblasts rather than cardiomyocytes. 

Due to the recent Piezo1 discovery, there is an obvious feeling of necessity to expand the knowledge in terms of its potential role within the cardiac context. However, despite the intention of this review, which is highly focused on cardiac mechanobiology, these channels, Piezo1 included, are also investigated both in other organs’ fibroblasts and in non-fibroblasts cell types.

## 6. Piezo1 Channel

Piezo1 is a recently discovered mechanosensitive or stretch-activated ion channel. Coste et al. in 2010 discovered the potential of this channel, expressed by several mammalian cells, and used to translate mechanical force into biological signals [90]. Piezo1 is a non-selective channel activated by pressure, hence the Greek word “Piezo”. It was observed to be permeable to Na^+^, K^+^, Ca^2+,^ and Mg^2+^ with a slight preference for Ca^2+^ [90]. Despite the in vitro chemical activation of Piezo1 by Yoda1 or Jedi1/2 [91,92], among the other mechanosensitive channels, Piezo channels are the only ones to be gated primarily by mechanical stimuli instead of chemical or physical ones [23]. In 2015, Yoda1 was first identified as a synthetic Piezo1-agonist compound that elicits Ca^2+^ flux selectively in Piezo1 in the absence of externally applied pressure [91]. Even if this agonist is the most used for channel studies purposes, in 2018, a novel set of Piezo1 chemical activators, termed Jedi (Jedi1 and Jedi2), was discovered [92]. While these offer the possibility to study the channel activation, there are currently very limited inhibitors that are specific for Piezo1 available. Non-specific antagonists of other mechanosensitive ion channels, such as streptomycin and spider peptide toxin (GsMTx4), do work for Piezo1 but not in isolation [93].

The Piezo family is composed of Piezo1 and Piezo2. They are both mechanically activated cation channels mainly located at the plasma membrane. However, Piezo1 was also detected in the endoplasmic reticulum, nucleus, and mitochondria [90,94]. They were found to be involved in many mechanotransduction pathways such as touch sensation, proprioception, nociception, vascular development, and breathing [95]. Up to now, Piezo2 has been considered to belong mainly to sensory neuron biology [96,97], while Piezo1 was shown to be expressed by many different cell types, also playing a vital role in the cardiovascular system [9]. However, recent data suggest that Piezo2 is also expressed in the myocardium [20,98,99], even if its role in the heart remains to be elucidated.

The Piezo1 structure is characterized by a three-blade propeller architecture [100] with an extracellular domain. The so-described structure consists of an in-plane dimension of about 200 Å and 140 Å in section [95]. The transmembrane helices have a pronounced bend culminating in a spherical dome, with an estimated dimension of 390 nm^2^ [101], projected into the cell, which is supposed to be responsible for the local membrane distortion even outside the channel perimeter [95,100,101]. A simplified representation is shown in Figure 2, which was adapted based on Haselwandter and co-workers’ illustrations [101]. Based on energy calculation, it is believed that the membrane deformation caused by this large protein leads to prefer a shape associated with the lowest energy. The work required to deform the membrane can be calculated from:$$G_{M}=\frac{1}{2}K_{b}\int {\left(c_{1}+c_{2}\right)}^{2}dA+\gamma \Delta A$$
where *K_b_* is the membrane bending modulus, γ is the membrane tension, *c*_1_ and *c*_2_ are the principal curvatures of the surface, and Δ*A* is the decrease in in-plane area when the membrane is deformed from its planar configuration [101]. As the equation suggests, it is possible to correlate the membrane deformation imposed by Piezo to only three physical properties (Figure 2). These parameters are the Piezo shape (basically its radius of curvature), the membrane bending modulus, and the membrane tension.

When tension is applied to the membrane, the energy is, in turn, minimized if Piezo flattens (rise in dome radius of curvature), favoring its opening state. Therefore, in relation to Figure 2, if the red curve is considered representative of the resting condition (closed state), the gradual tension application is described by the green followed by the blue curve, where the cell membrane appears almost completely flattened. For a detailed description of the energies involved, please refer to the recently published work by Haselwandter and co-workers [101]. In another recent article, Glogowska and collaborators anticipated that the Piezo1-mediated change in membrane curvature could, in turn, cause a change in the local lipid environment which influences other neighboring transmembrane proteins [102]. In their work, they show an increased TREK-1 current as a consequence of membrane cholesterol depletion in HEK293 cells lacking endogenous Piezo1. However, the same result was observed in control cells expressing Piezo1. Moreover, the current amplitude was then found to be enhanced as a consequence of the double effect of Piezo1 and cholesterol depletion, validating a possible common pathway of action for both. In addition, in their cell-attached patch clamp experiments, the authors always obtained a Piezo1 activation prior to TREK-1 opening. In conclusion, they addressed a local depletion of membrane cholesterol to the Piezo1 opening state which could be responsible for the TREK-1 prestress and gating kinetics modulation with a subsequent increase of its current amplitude.

Based on the aforementioned energetic discussion, force-induced membrane tension opens the channel allowing the permeation of cations [103]. However, since Piezo1 was discovered, two main models for its activation have been debated: force-through-lipid and force-through-filament models. Even if both models are based on changes in membrane tension, they intrinsically carry sophisticated differences in how these originate. The force-through-lipid model relies on the Piezo1 gating based only on local membrane lipid bilayer mechanical tension [104,105]. Even if channel activation was observed in the absence of cytoskeleton (e.g., experiments on membrane blebs) [103] favoring the force-through-lipid model, tethers connecting membrane and cytoskeleton were also observed to play a role in Piezo gating (force-through-filament model) [101,106,107]. This latter mechanism is nowadays getting even more attention, and recently, the tethered connection which links Piezo1 to the cytoskeleton was discovered. This one was found to reside within the E-cadherin/β-catenin/F-actin mechanotransduction complex [108] and provides for the long-range propagation of membrane tension perturbation. Therefore, due to the well-known ability of cells to exert acto-myosin-based endogenous traction forces, increases in local membrane tension can be generated even in the absence of external stimuli causing channel opening [108,109]. At the same time, a local membrane stimulus can be transmitted across the cell through cytoskeletal actin for the activation of far-placed Piezo1 channels. The upstream signaling responsible for the generation of traction forces that activate Piezo1 involves the phosphorylation of Myosin II by Myosin light chain kinase (MLCK). Moreover, since Ca^2+^ regulates MLCK itself, it likely represents the driving force for the feedback loop for which traction forces-induced Piezo1 activation is enhanced as a consequence of Piezo1-dependent calcium signaling [109]. Cell-generated traction forces are an absolute prerequisite for probing the stiffness of the ECM and regulating, in turn, cell signaling and function [106,109]. Therefore, these findings likely highlight the Piezo1 role for the downstream as well as the upstream signaling in sensing the external mechanical environment. Moreover, the Piezo1-dependent cytoskeletal remodeling was also confirmed by Vladislav I. Chubinskiy-Nadezhdin et al., who observed a Yoda1-dependent cell morphological change in an immortalized fibroblast cell line. In fact, 18 h of incubation with Yoda1 (30 μM) revealed an increase in cell area accompanied by actin stress fibers formation [110].

Another explored upstream Piezo1 role consists in the regulation of other mechanosensitive channels. In fact, it has been debated if the TRP channel family, as part of demonstrated mechanosensory systems, is actually sensible to membrane stretch. The answer finds a place in a recent article in which many TRP channels were investigated [111]. None of the considered channels allowed the group to observe a direct activation as a consequence of membrane stretch. Among the analyzed channels, TRPC6, TRPV1, and TRPV4, expressed by CFs, were considered. Therefore, TRP channels likely act downstream of other channels responsible for the mechanical stimulation signaling, such as Piezo1, as already proposed for TRPM4 [89,112] and TRPV4 channels [113,114]. Lastly, the role of Piezo1 as an upstream regulator of potassium-selective TREK-1 was highlighted [102], as previously mentioned.

All this evidence pushes toward a deep understanding of Piezo1 function due to its reasonable possible involvement as a key driver of many biological processes and cascades.

Since calcium entry regulates several cellular vital functions such as gene transcription, cell growth, differentiation, migration, and apoptosis [115,116,117], abnormal Ca^2+^ signaling can be related to several diseases in metabolism, neuron degeneration, immunity, and malignancy [118]. Therefore, the dysfunctional activity of channels responsible for calcium gating, like Piezo1, can represent the cause of several pathological conditions. Some evidence for this has been already highlighted, but other possible implications are still under investigation.

## 7. Piezo1 in Diseases

Loss-of-function Piezo1 mutations had been linked with Generalized Lymphatic Dysplasia [119], while gain-of-function mutations were found to cause dehydration of red blood cells within anemia [96,119]. Due to the recent discoveries, many studies are now investigating new possible genetic associations and, hopefully, future clarification about Piezo1’s role in pathologic human diseases and, perhaps, a role for pharmacological interventions targeting this channel in precision therapy.

Another field explored in terms of Piezo1 activity relates its role in cancer development and tumor progression. Curiously, based on the cancer type, Piezo1 was either found to be upregulated (in breast, gastric, prostate, and bladder cancers) or downregulated (as observed for lung cancer). For a detailed description of Piezo1 in these cancers, we refer the reader to [23]. This channel played a particularly critical role in glioma aggressiveness in the study by Xin Chen and coworkers [120]. They provided a comprehensive discussion about the underlying mechanism of this brain tumor, mainly based on a mechanical hypothesis. While the brain has a tissue stiffness of about 200 Pa in healthy conditions, the Young modulus is increased during tumor progression. They observed that the stiff mechanical environment induces Piezo1 mechanosensors to activate signaling pathways in favor of tumor aggression, accessed by increased tumor growth. In fact, they observed a Piezo1 upregulation in almost all types of glioma compared to normal brain tissue. In their work, they suggested a mechanism by which the tumor mechanical environment promotes the increased activation of Piezo1. Consequently, the calcium influx stimulates cell proliferation and ECM remodeling, enhancing once more brain tissue stiffness. In turn, the increased stiffness is responsible for Piezo1 upregulation, and the fate of this is a chronic feed-forward mechanism that promotes malignancy. Furthermore, by deleting Piezo1 in an animal model, they observed inhibition of tumor growth and prolonged animal survival, demonstrating the necessity of Piezo1 for aggressive tumor behavior.

The interconnection between some of these cancers and Piezo1 activity opened a new research field for future targeted therapies against this protein. Its unique biochemical characteristics make it a potential candidate as a new diagnostic biomarker in various other cancers (e.g., gastric, colorectal, prostate) as well as a treatment target [21,22]. Unfortunately, the absence of specific inhibitors against Piezo1 is limiting in this regard, as well as in improving our understanding of the physiological roles of this channel in terms of an untoward effect of inhibition on normal biology [23].

Moreover, a relevant Piezo1 role was also recently addressed in the regulation of renal fibrosis [121]. Xiaoduo Zhao and co-workers highlighted not only an upregulation of Piezo1 protein expression associated with renal fibrosis (also confirmed by TGF-β stimulation), but also a reduction in fibrotic markers such as fibronectin, collagen I, TGF-β, α-SMA by using the non-specific Piezo1 inhibitor GsMTx4. Despite the non-specificity of this one, the role of Piezo 1 was also studied by using the selective activator Yoda1. They observed that the Yoda1-stimulated Piezo1 activity is correlated to an increased expression of pro-fibrotic factors such as fibronectin and TGF-β. In addition, a Yoda1-dependent increased protein abundance of calpain2, a calcium-dependent protease, was observed. Meanwhile, Piezo1 small interfering RNA (siRNA) confirmed the role of the channel as an upstream regulator of calpain2. As well, the protein expression of talin1, which can be cleaved by calpain2 into an active form, was increased as a consequence of Yoda1. Because cleavage of talin1 by calpain2 is responsible for the increased affinity of integrin β_1_ through an inside-out mechanism [122,123], the protein expression of the latter was accessed, and its abundance was confirmed after Yoda1 treatment. Analogous results were also obtained by using stiff polymeric substrates which simulate a fibrotic environment; these were related to control soft ones, which mimic the healthy conditions. Finally, these results validated the relevant role of Piezo1 activation as an upstream activator of integrin β_1_ and, thus, following pro-fibrotic alterations. In fact, as proposed by Henderson et al. for liver fibrosis [68] and previously mentioned (see “Mechanical remodeling and paratensile effector”), α_v_β_1_ may be the major integrin involved in promoting the fibrotic character through the activation of TGF-β, which, in turn, drives both the deposition of the ECM [67,71,121] and Piezo1 overexpression [121]. These achievements forced us to look for possible Piezo1 implications also in cardiac diseases, and especially in cardiac fibrosis, highlighting possible new strategies for preserving cardiac health.

## 8. Piezo1 and the Heart

Piezo1 has been found to play a critical role in the outflow tract and aortic valve development [124,125]. Since its expression had been shown to be present in both cardiomyocytes and cardiac fibroblasts [96,126,127], possible correlations between this channel and diseases of the cardiovascular system are a natural next focus for investigation. Table 1 summarizes the current literature, reviewed below, about Piezo1 dysfunctional regulation in cardiac cells (myocytes and fibroblasts) and our preliminary hypothesis about its beneficial targeting in fibroblasts.

First, it was observed that stretch-induced calcium influx in cardiomyocytes is mediated by Piezo1 [19]. Moreover, Fan Jiang et al. showed that, by deleting Piezo1 from mice cardiomyocytes, both sarcoplasmic reticulum Ca^2+^ content and spontaneous Ca^2+^ activity were reduced. In addition, they observed that Piezo1 knockdown produces a larger heart with dilated left ventricles and enhanced fibrosis associated with the development of cardiomyopathy. On the other hand, cardiac-specific overexpression of Piezo1 was also related to arrhythmias and heart failure [19].

In this way, these investigators highlighted the critical role of Piezo1 in maintaining homeostatic Ca^2+^ signaling in cardiomyocytes, an absolute prerequisite for proper heart function. In another report, Yuhao Zhang et al. [20] detected elevated Piezo1 expression in the pressure-load hypertrophy and in isolated hypertrophic cardiomyocytes, as well. Moreover, they demonstrated an attenuated progression of the pathological cardiac hypertrophy in cardiac-specific Piezo1 knockdown mice and in cardiomyocytes, but also a reduction in fibrotic remodeling. These achievements were recently confirmed by Ze-Yan Yu and collaborators [112]. In their article, they provide evidence about Piezo1 and TRPM4 physical interaction, which is responsible for the activation of the signaling cascade resulting in pathological hypertrophy. The same group had previously demonstrated the role of TRPM4 as a positive regulator of left ventricular hypertrophy induced by pressure overload [89]; however, only recently were able to address the TRPM4 channel activity to a downstream effect of Piezo1. Therefore, in this last work [112], Piezo1 was confirmed as the primary mechanotransducer that initiates the pressure overload response via TRPM4.

Despite several studies about Piezo1 in the cardiomyocyte [19,20,112,128], however, only recently have reports been started to focus on its role in cardiac fibroblasts. Many efforts are now focusing on possible fibroblasts Piezo1-mediated heart dysfunctions, and due to the well-known role of these cells in fibrosis, it is supposed that promising and helpful new Piezo1-based mechanisms could be revealed.

In one report [130], Piezo1 dysfunctional regulation was investigated within atrial fibrillation. It was observed that in non-passaged right atrial fibroblasts from atrial fibrillation (AF), patients’ Piezo1 expression and activity were higher compared to those patients in sinus rhythm (SR). However, when passaged fibroblasts were analyzed, it was observed that SR cells become more “AF-like” with the absence of a significant difference in Piezo1 between the two conditions [130]. This important achievement highlights the fibroblasts Piezo1 role in this kind of supraventricular arrhythmia. Moreover, in other works, Piezo1 was linked to cytokines secretion with consequent adverse cardiac outcomes [127,132]. It was demonstrated that Yoda1 10 µM treatment for 4 h leads to an increased mRNA expression of TGF-β [132], while 0.5–10 µM for 24 h was found able to cause the same effect on IL-6 mRNA [127]. This increased IL-6, which was reported by the same group to contribute to cardiac hypertrophy [131], was reduced in fibroblasts-transfected with Piezo1-specific siRNA, highlighting the pivotal role of this channel. By accessing differences in activity of several kinase families following Yoda1 treatment, they observed a Piezo1 upstream effector. However, among the selective inhibitors, only p38 mitogen-activated protein kinase (MAPK) inhibition was able to reduce the Yoda1-induced increase in IL-6 expression [127]. The requirement of Piezo1 for the increased IL-6 secretion was discussed, showing the Yoda1-induced increased p38 phosphorylation (activation), while Piezo1-specific siRNA reduced it. Moreover, Nicola M. Blythe et al. [127] also achieved the conclusion that the substrate stiffness can modulate accordingly the IL-6 content. In fact, they demonstrated the Piezo1-siRNA ability in reducing the basal IL-6 expression when fibroblasts are grown on soft substrate but not on rigid ones. This finding highlights once more the previously mentioned (see the section “Piezo1 channel”) potential role of Piezo1 in signaling substrate properties, a feature of high interest in the fibrotic remodeling field. In addition, this agrees with Xiaoduo Zhao et al.’s finding that Piezo1-dependent increased β_1_ activation on the stiffer environment in the renal context [121]. Alongside this, it is opportune to remember that, related to other organ fibrosis (kidney, lung, liver), α_v_β_1_ and traction forces (which are favored by Piezo1-induced Myosin II phosphorylation by MLCK (see Section 6)) were found to play a pivotal role in TGF-β activation with subsequent ECM and Piezo1 protein expression remodeling [67,68,71,121]. Thus, if these statements are translated to the cardiac case, they may be enough to consider the Piezo1 role critical for the cardiac fibrotic progression.

In fact, based on these stiffness-mediated cytokines regulation through Piezo1 activity and its positive feedback on ECM production, the progression of pathologic fibrosis may suggest new possibilities for its targeted disruption in a therapeutic sense.

Specifically, Piezo1 involvement in cardiac fibrosis was already taken into consideration by Fiona Bartoli et al. in a very recent article [126]. Their work shows that a global gain-of-function Piezo1 mutation in mice is responsible for the amplificated calcium flow and both the overstimulated downstream p38 signaling and IL-6 secretion in cardiac fibroblasts. This was correlated with cardiomyocyte hypertrophy, as already mentioned in this work, but also cardiac fibrosis, confirmed by increased mRNA Col3a1 (type III collagen) gene expression.

Therefore, it can be summarized that Piezo1 activation may guide both autocrine and paracrine pathways representing a key driving force for cardiac remodeling and myofibroblasts persistence. In fact, higher traction forces exerted by myofibroblasts may increase the channel opening as a consequence of membrane tension. Then, the calcium-activated p38 MAPK mediates the rise in IL-6 secretion with subsequent adverse paracrine effects on cardiomyocytes as well as an autocrine behavior on still quiescent fibroblasts and on already existent myofibroblasts by enhancing their features. Moreover, due to calcium-stimulated TGF-β secretion and ECM protein synthesis, paratensile effectors also occur. In fact, new myofibroblasts might result from myofibroblasts–fibroblasts cross-talk through the stiffened fibrous matrix, whereas the stiffer matrix represents the source for traction forces generation for restarting the process loop. In this last mechanism, we also remember the interconnection between TGF-β and ECM, in which excessive protein deposition occurs as a consequence of the cytokine activation. Therefore, TGF-β contributes to enhancing the matrix stiffness and, thus, not only acts as a direct paracrine/autocrine effector but also as an indirect paratensile one (Figure 3). 

Lastly, the role of Piezo1 in cardiac fibroblasts stiffness regulation was proposed in a recent article by Ramona Emig et al. [129]. Based on a more mechanical approach, they observed an increase in cell stiffness in Piezo1-overexpressed human atrial fibroblasts while Piezo1-siRNA showed a reduction in the cell’s Young’s Modulus. In this way, the role of Piezo1 in regulating cell stiffness was highlighted. Possibly, the increase in stiffness may be related to the increased Ca^2+^ flow with the consequential reorganization of the cytoskeleton [118].

The work by Ramona Emig and collaborators demonstrates, indeed, that this cytoskeletal reorganization really occurs. In fact, Piezo1-overexpressed cells exhibited a higher cell area covered by thicker actin bundles compared to control, which is in agreement with Chubinskiy-Nadezhdin et al.’s findings previously mentioned [110]. Even if, in recent years, Piezo2’s role was addressed to an upstream regulator of RhoA, which in turn, through signaling cascades, is involved in stress fibers formation [133], the requirement of Piezo1 for the actin remodeling needed in the myofibroblasts transition was demonstrated through its knockdown by Longwei Liu and colleagues [66]. Finally, Piezo1’s role in substrates stiffness adaptation, a well-known cell feature [11,12], was stated. In fact, cells grown on stiff substrates had been observed to acquire increased stiffness while Piezo1 knockdown prevented the same cell adaptation [129]. In addition, Piezo1-induced cell stiffness (PiCS) was observed to be transferrable between neighboring cells, for example, from a Piezo1-overexpressing cell to a non-transfected control one, which likely occurred as a consequence of IL-6 autocrine signaling and abolished by neutralizing IL-6. If we translate these results into our approach based on the myofibroblasts’ Piezo1-mediated mechanism for fibrosis progression, we can consider myofibroblasts as the cells characterized by increased Piezo1 expression and activity. Thus, the Piezo1-mediated increase in IL-6, used by myofibroblasts themselves for their maintenance, can also influence adjacent fibroblasts by favoring their mechanical properties regulation and tissue stiffness adaptation, promoting the pathologic character (Figure 4).

Therefore, it can be concluded that these results might be extremely interesting if applied to cardiac remodeling. In fact, it implicates a self-sustaining cell-mediated mechanism by which the progressive ECM stiffening begets enhanced stiffening through continuous cytoskeletal remodeling and myofibroblasts recruitment, all of which could be driven/enhanced by Piezo1. For these reasons, it may represent a potential therapeutic target to break this feed-forward cycle.

## 9. Piezo1: Potential Target in Cardiac Fibrosis

Calcium flow is known to be enhanced when fibroblasts are cultured on stiff substrates [134]. This may be coupled with the increased tension exerted on the membrane by cell traction forces which favor the opening of Piezo1 [103]. Thereafter, the increased Ca^2+^ influx will promote the actin reorganization, which is complemented by thicker actin bundles [129], and ECM remodeling [115], forcing the phenoconversion of fibroblasts into myofibroblasts. Once established, the paratensile, autocrine, and paracrine signaling are capable of enhancing fibrotic remodeling in a similar manner as seen with tumor progression [120]. Based on Piezo1 recent discoveries, such as Piezo1-dependent (i) cell stiffness regulation, (ii) ECM remodeling, (iii) adaptation to substrate, and (iv) cytokines secretion, this channel emerges as a possible key player in fibrosis, having a pivotal role in orchestrating all of the maladaptive fibroblasts features. The importance of such an upstream position for Piezo1 is driving several studies that target its activity for future successful antifibrotic therapies rather than other proposed interventions, such as the reduction in ECM stiffness [135].

On this topic, it was shown that fibroblasts possess a mechanical memory of past experienced environments, which would antagonize an approach to restoring physiological heart function solely by focusing on the ECM. Balestrini et al. verified this behavior in lung fibroblasts, showing that cells cultured under stiff pathological conditions (substrates which mimic the fibrotic elasticity) are able to continuously exhibit typical features of myofibroblasts (e.g., *α*-SMA, contractility, and proliferation) even after being shifted to soft substrates [136]. Meanwhile, investigating the nature of mechanical memory, Samila Nasrollahi et al. observed that, by monitoring the nuclear accumulation of yes-associated protein (YAP), even epithelial cells are able to possess a memory of past conditions. They observed that depletion of YAP is able to almost eliminate the memory, suggesting a key mechanism for storing mechanical memory into the subcellular YAP localization [137]. The article published by Medha M. Pathak et al., based on a study of human neural stem/progenitor cells (hNSPCs), links YAP to the discussion on Piezo1 [138]. They observed that Piezo1 knockdown interferes with the YAP mechanoresponses by reducing its nuclear localization even on the rigid substrate where it usually occurs. The observed evidence allowed the group to relate the YAP activity to the downstream effect of Piezo1. Since the mechano-signaling through YAP also occurs in fibroblasts [139], this observation provides promising ideas about the potential involvement of Piezo1 in their mechanical memory mechanism as well. Considering these findings reinforces the concern that focusing on cardiac remodeling purely by reducing the ECM stiffness [18,140] is likely to be insufficient.

Therefore, this validates the necessity to develop new successful approaches for targeting elements involved on the upstream side. Based on the deep analysis provided here, consisting of several pieces of evidence about a potentially pivotal role of Piezo1 in cardiac fibrosis, this novel protein seems to possess all the characteristics for being used as a suitable future target in antifibrotic precision therapies. However, although a very tantalizing prospect, these observations remain in the early stages, and additional studies need to be carried out for a deeper understanding of the role of this channel.

Nevertheless, at least for now, it seems that only after targeting Piezo1 and interrupting the feed-forward chain which drives the chronic progression will the mechanical properties of the myocardium be restored, resulting in proper cardiac function.

## 10. Conclusions

Cardiac fibrosis is a hallmark of several cardiac diseases (e.g., MI, aging, hypertension, and hypertrophic cardiomyopathy) with a high incidence of cardiac morbidity and mortality. Even if clinical analyses are nowadays employed for diagnostic purposes, these are mostly based on indirect measurements of fibrosis [141]. Conversely, direct assessment of cardiac fibrosis requires invasive procedures, especially given the diffuse nature of the disease process [141,142,143]. At the same time, once diagnosed, the challenge remains in its therapeutic reversal [143]. The poorly understood pathophysiological mechanisms established in the fibrotic process limit, to date, its successful treatment [18]. Even as some heart failure treatments have been shown to reduce significantly cardiac fibrosis, specific antifibrotic drugs are still absent. Many researchers are now focusing on the possible inhibition of pro-fibrotic signaling and the activation of antifibrotic pathways [1]. Promising results have been achieved, but, at least for now, they are limited to animal models or in vitro analyses [34,38,144,145,146]. Beneficial effects of some antifibrotic therapies have been observed in clinical studies on humans, but the population investigated is still too small [29,147] and the possibility of translating animal results to human patients is hampered by the obvious physiologic and genetic differences between them [29,148].

Due to the severity of illnesses seen with the progressive fibrosis of a number of organs (lung, liver, heart, kidney), it is an absolute priority to investigate new, promising biomarkers and potentially beneficial clinical therapies.

As discussed for tumor cases and as already proposed for renal tumors [21,121,149], Piezo1 could also be, at least, a biomarker for discriminating the aggressiveness of cardiac fibrosis and, at most, a beneficial therapeutic target. Moreover, the possibility of restoring its physiological activity may interrupt the positive feedback in which its increased activity promotes ECM remodeling in a way in which its stiffness-induced stimulated activity is again enhanced. In addition, its involvement in cytokines regulation, the mechanism by which it affects healthy fibroblasts even distant from the injured zone, may open the possibility of slowing down the organ-wide reactive fibrosis. However, even with the promise of the reported observations on the benefit of inhibiting Piezo1 activity, more studies need to be carried out starting from in vitro experiments. Mimicking physiological and pathophysiological stiffness conditions is easily achieved by using biocompatible substrates of different materials [150] (e.g., silicone elastomer [36,39], polyacrylamide [46], alginate [151]). In addition, by using already known biological strategies for altering Piezo1 expression/activity, such as siRNA for Piezo1 silencing and specific agonists like Yoda1, important results can still be accomplished. Despite the cited differences with humans, animal models provide a wide field of exploration for these purposes. In fact, direct studies on pathological fibrotic animal hearts could well uncover hidden mechanisms not well appreciable in cultured cell manipulation. We believe that, based on the preclinical work reported here, Piezo1 could play an important role in cardiac fibroblasts biology and offers a unique opportunity to modify the maladaptive cardiac remodeling in response to injury, ultimately leading to heart failure.

## Figures and Tables

**Figure 1 ijms-23-08065-f001:**
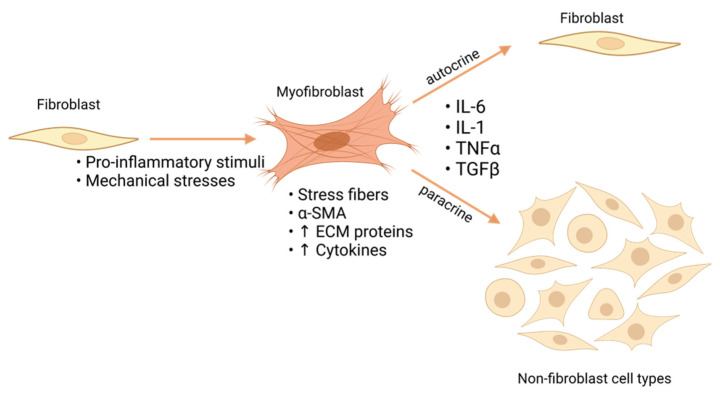
Fibroblasts activation mechanism. Pro-inflammatory signaling and mechanical stress after cardiac injury promote the activation of fibroblasts into myofibroblasts. Myofibroblasts, characterized by actin stress fibers, α-SMA, increased cytokines (TGF-β, IL-6, IL-1, TNFα), and ECM protein secretion, are responsible for both paracrine and autocrine effects on adjacent cells. α-smooth muscle actin (α-SMA); Transforming growth factor β (TGF-β); Interleukins (IL-6, IL-1); tumor necrosis factor α (TNFα); extracellular matrix (ECM). Created with Biorender.com (accessed on 15 June 2022).

**Figure 2 ijms-23-08065-f002:**
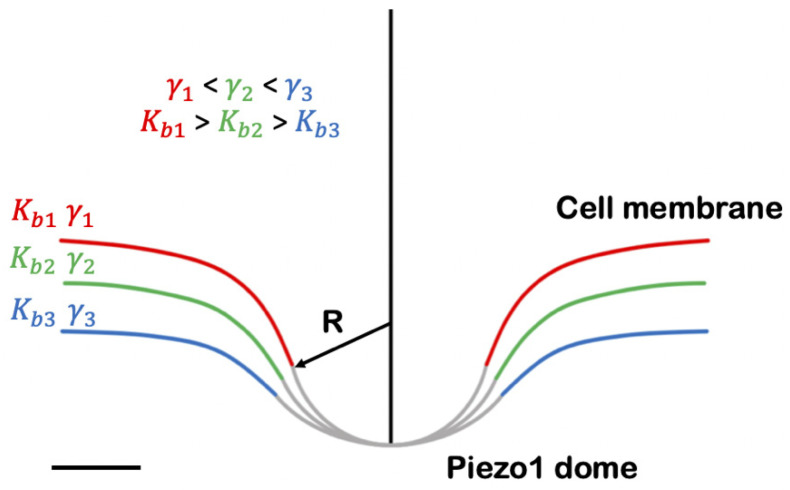
Section view of Piezo dome (gray) and deformation imposed on the cell membrane (red, green, blue). The three physical parameters and their relative relationships involved in the regulation of membrane shape are reported: R (radius of curvature of the dome), *K_b_* (membrane bending modulus), and γ (membrane tension). Scale bar: 10 nm.

**Figure 3 ijms-23-08065-f003:**
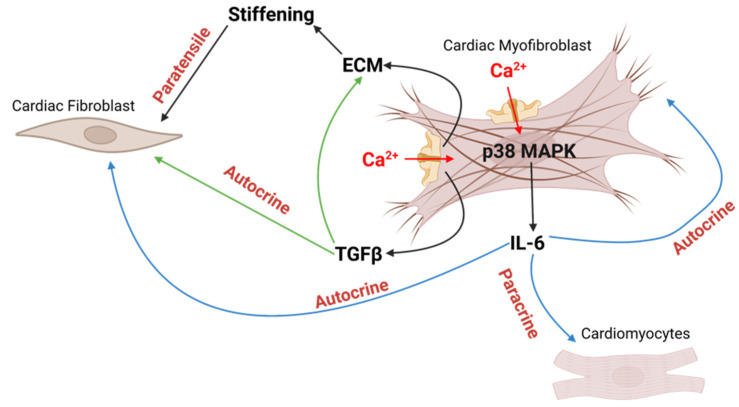
Piezo1-induced autocrine, paracrine, and paratensile effectors in activated fibroblasts. Increased Piezo1 activity can activate the p38 MAPK pathway with enhanced IL-6 secretion. IL-6 acts as a paracrine effector on cardiomyocytes, as well as an autocrine effector for fibroblasts and myofibroblasts themselves. Calcium-stimulated ECM synthesis culminates in the activation of new fibroblasts through a paratensile manner. However, Piezo1-stimulated TGF-β activation enhances ECM synthesis and acts not only as an autocrine effector but also as an indirect paratensile one. Created with Biorender.com (accessed on 15 June 2022).

**Figure 4 ijms-23-08065-f004:**
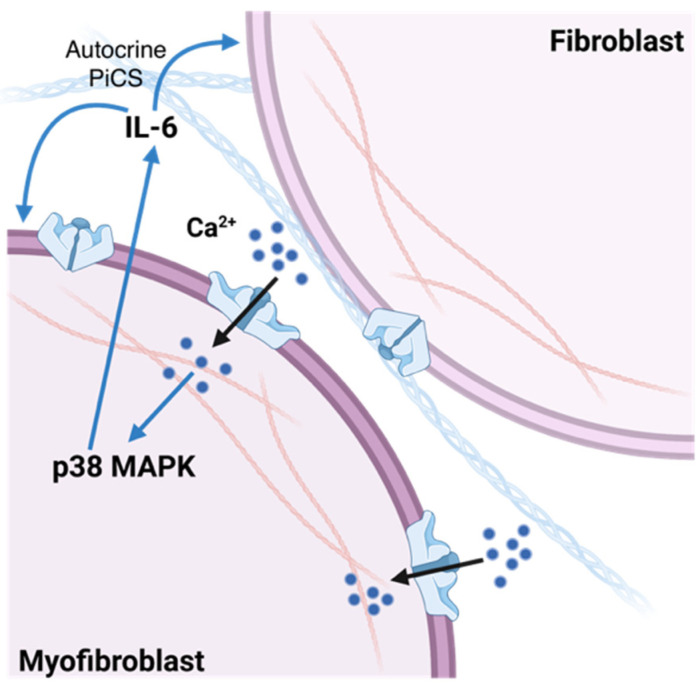
Piezo1-induced cell stiffness (PiCS). Calcium-dependent increased stiffness is mediated by the p38 MAPK pathway and requires IL-6. Adapted from “ECM (2 cell interaction”) by BioRender.com (accessed on 15 June 2022). Retrieved from https://app.biorender.com/biorender-templates.

**Table 1 ijms-23-08065-t001:** Overview of Piezo1 dysfunctional regulation in cardiac cells (myocytes and fibroblasts) and related cellular/clinical features. * Refers to our hypothesis about Piezo1 beneficial targeting in fibroblasts discussed in the present work.

Cell Type	Expression	Cellular/Clinical Features	References
**Cardiomyocytes**	knockdown	Reduced sarcoplasmic reticulum Ca^2+^ content and spontaneous Ca^2+^ activity Larger heart with dilated left ventricles and fibrosis Attenuated progression of the pathological hypertrophy	[19,20,112]
	overexpression	Arrhythmias Heart failure Hypertrophy Fibrosis	[19,20,112,128]
**Cardiac Fibroblasts**	knockdown	Inhibited cardiac fibroblasts activation Reduced cytokines release Prevented cell adaptation to substrate feature Prevented cardiac fibrotic invasive character *	[66,127,129] * Our hypothesis discussed in the present work
	overexpression	Atrial fibrillation Increased cytokines release Hypertrophy via IL-6 secretion Increased cell stiffness Favored myofibroblast phenotype Enhanced ECM accumulation Fast recruitment of new myofibroblasts * Increased fibrotic remodeling *	[126,127,129,130,131,132] * Our hypothesis discussed in the present work

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
