# Peer review of "Piezo1 Channel as a Potential Target for Hindering Cardiac Fibrotic Remodeling"

_ijms, 2022, doi:10.3390/ijms23158065_

Round 1

Reviewer 1 Report

This is a review of the Piezo1 channel as a potential target for suppressing cardiac fibrotic remodeling. It’s a very informative review that cites many recent papers.

However, unfortunately, there are some problems as it is.

Figures 1, 3 and 4 are very similar, so the authors need to devise a little more, and I think it is better to focus on what the authors want to say. Fig.2 is hard to understand. Please check if Citation 99 is appropriate. Citation 127 also needs to be described accurately.

Author Response

Review 1

Figures 1, 3 and 4 are very similar, so the authors need to devise a little more, and I think it is better to focus on what the authors want to say. Fig. 2 is hard to understand. Please check if Citation 99 is appropriate. Citation 127 also needs to be described accurately.

We carefully considered the Reviewer comments and adjusted the paper accordingly:

  • Figure 2 was adjusted for a better comprehension, also adding a better explanation of it within the text and capture.
  • Citations 99 and 127 have been checked and opportunely discussed in the text.

However, we found difficult to change figure 1, 3 and 4, while at the same time, maintaining a clear message to the readers. Therefore, also taking into consideration the positive comment of second Reviewer (The paper is well organized and written with excellent figures), we suggest to leave these figures unchanged. Our motivations are discussed below:

  • Figure 1 is a schematic illustration of historically known causes and consequences of fibroblasts activation which are not correlated with Piezo1;
  • Figure 3 refers to autocrine and paracrine effectors mentioned in Figure 1 but also taking into consideration Piezo1’s role. The multiple meanings enclosed in the figure consist in:
    • IL-6 release as a consequence of Piezo1-dependent p38 phosphorylation;
    • Paratensile effector as a consequence of Piezo1-dependent ECM accumulation;
    • Indirect paratensile effector as a consequence of Piezo1-dependent TGFb activation which promote ECM remodeling.
  • Figure 4 refers to IL-6 secretion based on the Piezo1-mediated mechanism reported in Figure 3 (through p38 MAPK pathway), which is required for both augmented cell stiffness in Piezo1 overexpressed cells (which we suppose are myofibroblasts) and induced cell stiffening in neighboring cells (quiescent fibroblasts).

Therefore, even if some parts are repeated in the figures, we find these necessary for introducing additional information each time.

We thank again the Reviewer for his/her consideration and suggestions, and hope to have fulfilled all requests.

Reviewer 2 Report

In the paper  "Piezo1 Channel as a Potential Target for Hindering Cardiac Fibrotic Remodeling" the authors summarized recent discoveries related to the role of the mechanosensitive ion channel Piezo1 in several diseases, especially in regulating tumor progression, and how this can be compared with cardiac mechanobiology. 

The paper is well organized and written with excellent figures

Minor points 

1. Please add some tables for summarized the role of the piezo 1 channel in the pathophysiology of heart fibrosis as well as the role of piezo 1 as potential target in cardiac fibrosis

Author Response

Review 2

In the paper  "Piezo1 Channel as a Potential Target for Hindering Cardiac Fibrotic Remodeling" the authors summarized recent discoveries related to the role of the mechanosensitive ion channel Piezo1 in several diseases, especially in regulating tumor progression, and how this can be compared with cardiac mechanobiology. 

The paper is well organized and written with excellent figures

Minor points 

  1. Please add some tables for summarized the role of the piezo 1 channel in the pathophysiology of heart fibrosis as well as the role of piezo 1 as potential target in cardiac fibrosis

A summarizing table was added to the main text for a better comprehension of the discussion about the role of Piezo1 in the heart, in the particular in fibrosis, which is our focus.

We thank again the Reviewer for his/her consideration and suggestions, and hope to have fulfilled all requests.